# An empirical analysis of the impact of Chinese government investment on high-quality economic development——A study based on spatial Dubin model

**Ming Wang**[1], **Weiming Liu**[2]*

**1** Business School, Jiangsu Open University, Nanjing, Jiangsu, China, **2** School of Shipping and Economics, Guangzhou Institute of Navigation, Guangzhou, Guangdong, China

* yjzxwm@163.com

**Data Availability Statement:** The basic data used in this paper are basically derived from public information such as China Statistical Yearbook,

## Abstract

China's economy has shifted from the stage of high-speed growth to the stage of high-quality development. This paper selects panel data of 30 provinces, municipalities directly under the Central Government and autonomous regions (except Tibet) in mainland China with the time dimension of 2006–2016, selects indicators from five dimensions of innovation, coordination, greenness, openness and sharing, uses principal component analysis to construct a system of indicators of high-quality economic development, and uses the spatial Durbin model to empirically the impact of government investment behavior in the three industries on the regional economic high-quality development was studied. The results show that: first, there is a significant positive spatial correlation between China's economic quality development. Second, government investment in different industries has different effects on the quality development of the economy. Investment in the tertiary industry can boost employment, raise income level, promote economic quality development, and enhance sharing, coordination and innovation; investment in the secondary industry can enhance innovation, but because of the "crowding-out effect", it will reduce openness and sharing, and is not conducive to promoting economic quality development; investment in the primary industry can Investment in the primary industry can significantly improve coordination, but it will reduce innovation, greenness, openness and sharing, which will also have a negative impact on the overall high-quality economic development. Third, there is a significant spatial spillover effect of government investment behavior on regional economic quality development, and the economic quality development of this region will have an impact on neighboring regions. Finally, based on the research results, this paper puts forward countermeasure suggestions for promoting high-quality economic development.

## 1. Introduction

Since the reform and opening up, China has made remarkable achievements in economic development. It has become the world's second largest economy in just over 30 years. From

China Industrial Statistical Yearbook, China Fixed Assets Statistical Yearbook, China Tertiary Industry Statistical Yearbook, New China 60 Years Statistical Data Compilation, and the statistical yearbooks of various provinces and cities in the past years, etc. A few missing data were estimated by interpolation, regression analysis and other methods. Public information can be found from the China Knowledge Network China's economic and social big data research platform at https://data.cnki.net/. In addition, the China Statistical Yearbook can be found on the official website of the National Bureau of Statistics of China at http://www.stats.gov.cn/tjsj/ndsj/. Provincial and municipal statistical yearbooks can be found on the official websites of provincial and municipal statistical bureaus, with the exception of Xinjiang, the links to other provinces, cities and autonomous regions are as follows: China Statistical Yearbook: https://data.cnki.net/yearBook/single?id=N2022110021 China Industrial Statistical Yearbook: https://data.cnki.net/yearBook/single?id=N2022010304 China Science and Technology Statistical Yearbook: https://data.cnki.net/yearBook/single?id=N2022010277 China Population and Employment Statistics Yearbook: https://data.cnki.net/yearBook/single?id=N2022040097 China Environmental Statistical Yearbook: https://data.cnki.net/yearBook/single?id=N2022030234 China Energy Statistics Yearbook: https://data.cnki.net/yearBook/single?id=N2022030234 New China 60 years of statistical data compilation: https://data.cnki.net/yearBook/single?id=N2010042091 China Trade and Foreign Economic Statistics Yearbook: https://data.cnki.net/yearBook/single?id=N2022010261 China Labor Statistics Yearbook: https://data.cnki.net/yearBook/single?id=N2022020102 China Real Estate Statistical Yearbook: https://data.cnki.net/yearBook/single?id=N2022010276 China Construction Industry Statistical Yearbook: https://data.cnki.net/yearBook/single?id=N2021120002 China Financial Yearbook: https://data.cnki.net/yearBook/single?id=N2022040012 China Regional Economic Statistics Yearbook: https://data.cnki.net/yearBook/single?id=N2015070200 In addition, the China Statistical Yearbook can be accessed on the official website of the National Bureau of Statistics of China at http://www.stats.gov.cn/tjsj/ndsj/. Provincial and municipal statistical yearbooks can be found on the official websites of provincial and municipal statistical bureaus, with the exception of Xinjiang, the links to other provinces, cities and autonomous regions are as follows: Beijing Statistical Yearbook: http://tjj.beijing.gov.cn/tjsj_31433/ Tianjin Statistical Yearbook: https://stats.tj.gov.cn/tjsj_52032/tjnj/ Hebei Economic Yearbook: http://www.hetj.gov.cn/

1978 to 2020, the scale of China's economy continued to expand, with the actual GDP increasing from 367.87 billion yuan to 14.7213 trillion yuan, and the average annual growth rate of 9.18%. The level of economic development has been continuously improved, and the actual per capita GDP has increased from 385 yuan to 10,493.14 yuan, with an average annual growth rate of about 8.19%. With the improving of the level of urbanization continuously, the urbanization rate has increased from 17.92% to 63.89%. With the optimizing of the industrial structure continuously, the ratio of the three industries has changed from 27.7:47.7:24.6 to 7.7:37.8:54.5, and the contribution of high-tech industries to economic development has been continuously improved. However, after gaining these achievements, China has also paid a huge price, and many problems have become increasingly prominent, such as the uncoordinated industrial structure, the widening income gap between urban and rural areas, unbalanced regional development, low resource utilization efficiency, serious environmental pollution, and so on.

Since 2019, the COVID-19 has ravaged the world and has had a huge impact on world economic development, and it has profoundly affected and changed global and regional sustainability issues. As the second largest economy in the world, it is inevitable that China's economic development will be affected. On the one hand, the economic growth rate has decreased, with China's GDP growth rate of 6.0% in 2019 before the outbreak and 2.3% in 2020 due to the impact of the new crown outbreak. On the other hand, the new crown epidemic has different impacts on China's high-quality economic development in different ways, such as the new crown epidemic can help save energy and reduce carbon in the short term and have a positive impact on air, but in the long term, this impact is not sustainable (Wang, 2020) [1].

In recent years, the development of China's economic has entered a new normal. It needs to promote the transformation of economic development from Chinese speed to Chinese quality in accordance with the characteristics of the new era, and the requirements of high-quality development. In 2015, the Government of China put forward five development concepts of "innovation, coordination, greenness, openness and sharing", which embodies China's development ideas, development direction and development focus for a period of time in the future. In 2017, The President of China pointed out: "China's economy has shifted from a stage of high-speed growth to a stage of high-quality development, and it is in a critical period of transforming the mode of economic development, optimizing the industrial structure, and transforming the driving force of growth." With the increasing marketization in China, the role of the government in economic development is gradually changing and gradually transforming to a service-oriented government. In the context of the new era, it is of great theoretical and practical significance to study the role of government investment behavior on high-quality economic development to promote China's economy to achieve high-quality development.

## 2. Literature review

The academic community has a long history of research on high-quality economic development. Research on high-quality economic development has focused on two aspects: connotation and evaluation system. From the perspective of the concept of high quality economic development, the connotation of high quality economic development includes the improvement of the quality of the population, the improvement of the quality of the environment, the good financial situation and the improvement of production efficiency [2–6].Different scholars possess different views on the assessment system of high quality economic development, and there are generally two ideas in narrow and broad sense. One of the most widely thought of narrowly is the use of total factor productivity as a measure of quality economic

hetj/tjsj/jjnj/ Shanxi Statistical Yearbook: http://tjj. shanxi.gov.cn/tjsj/ Inner Mongolia Statistical Yearbook: http://tj.nmg.gov.cn/datashow/pubmgr/ publishmanage.htm?m=queryPubData&procode= 0003 Liaoning Statistical Yearbook: https://tjj.ln. gov.cn/tjj/tjxx/xxcx/index.shtml Jilin Statistical Yearbook: http://tjj.jl.gov.cn/tjsj/tjnj/ Heilongjiang Statistical Yearbook: http://tjj.hlj.gov.cn/tjj/ c106782/common_zfxxgk.shtml Shanghai Statistical Yearbook: https://tjj.sh.gov.cn/tjnj/index. html Jiangsu Statistical Yearbook: http://tj.jiangsu. gov.cn/col/col87172/index.html Zhejiang Statistical Yearbook: http://tjj.zj.gov.cn/col/col1525563/index. html Anhui Statistical Yearbook: http://tjj.ah.gov.cn/ ssah/qwfbjd/tjnj/index.html Fujian Statistical Yearbook: https://tjj.fujian.gov.cn/xxgk/ndsj/ Jiangxi Statistical Yearbook: http://tjj.jiangxi.gov.cn/ col/col38595/index.html Shandong Statistical Yearbook: http://tjj.shandong.gov.cn/col/col6279/ index.html Henan Statistical Yearbook: https://tjj. henan.gov.cn/tjfw/tjcbw/tjnj/ Hubei Statistical Yearbook: http://tjj.hubei.gov.cn/tjsj/sjkscx/tjnj/ qstjnj/ Hunan Statistical Yearbook: http://tjj.hunan. gov.cn/hntj/tjsj/tjnj/index.html Guangdong Statistical Yearbook: http://stats.gd.gov.cn/gdtjnj/ Guangxi Statistical Yearbook: http://tjj.gxzf.gov.cn/ tjsj/tjnj/ Hainan Statistical Yearbook: http://stats. hainan.gov.cn/tjj/tjsu/ndsj/ Chongqing Statistical Yearbook: http://tjj.cq.gov.cn/zwgk_233/tjnj/ Sichuan Statistical Yearbook: http://tjj.sc.gov.cn/ scstjj/c105855/nj.shtml Guizhou Statistical Yearbook: http://stjj.guizhou.gov.cn/tjsj_35719/ sjcx_35720/gztjnj_40112/tjnj2018/ Yunnan Statistical Yearbook: http://stats.yn.gov.cn/tjsj/tjnj/ index.html Shaanxi Statistical Yearbook: http://tjj. shaanxi.gov.cn/tjsj/ndsj/tjnj/ Gansu Statistical Yearbook: http://tjj.gansu.gov.cn/tjj/c109464/info_ disp.shtml Qinghai Statistical Yearbook: http://tjj. qinghai.gov.cn/tjData/qhtjnj/ Ningxia Statistical Yearbook: http://nxdata.com.cn/publish.htm?m= getMorePublish&bc=A01&cn=G01.

**Funding:** The paper was funded by the National Natural Science Foundation of China under the project 71763011, Science and Technology Project of Education Department of Jiangxi Province of China under the project GJJ200519, and Humanities and Social Sciences Research Planning Project of Universities in Jiangxi Province of China under the project JJ19104. The funders had no role in study design, data collection and analysis, decision to publish, or preparation of the manuscript.

**Competing interests:** The authors have declared that no competing interests exist.

development [7–11]. From a broad perspective, the economic quality development index system is mainly constructed from a multidimensional perspective. First, based on the well-being of people's livelihood, indicators are selected to examine the quality of economic growth in the areas of material living standards, political rights, education levels, life expectancy, health status and income disparity [12,13]. Second, from the supply, demand, efficiency, operation, openness and other aspects of the construction of high-quality economic development indicators system [14]. Third, based on the characteristics of the new economic normal, select indicators from three aspects of development mode transformation, economic structure optimization and growth momentum transformation to build a high-quality development evaluation index system [15]. Fourth, select indicators from both input and output perspectives to build an evaluation system for high-quality economic development [16,17]. Fifth, Zhan Xinyu et al. (2016) [18] and Shan Qinqin (2022) [19] analyze the construction of economic growth quality system with the five development concepts as a guide [18–20].

In addition to the research on the concept of high-quality economic development and the evaluation index system, the research on its influencing factors has also become a hot topic of research. The development of green finance, industrial agglomeration development, upgrading of technology, development of digital economy and implementation of environmental regulation can effectively promote high-quality economic development, mainly in the following aspects: green finance can have a positive impact on ecological environment, economic efficiency and economic structure; industrial agglomeration development can now improve production efficiency and reduce production costs; development of digital economy can effectively promote industrial transformation and upgrading; implementation of environmental regulation can significantly reduce energy use and reduce carbon and other pollutant emissions; upgrading of technology can significantly reduce energy use and reduce carbon and other pollutant emissions. The implementation of environmental regulations can significantly reduce energy use and carbon and other pollutant emissions; the improvement of technology level has a catalytic effect on the development of green finance, industrial agglomeration development, the improvement of technology level, the development of digital economy and the implementation of environmental regulations, thus promoting high-quality economic development [21–30].

Regarding the relationship between government investment behavior and high-quality economic development, relatively few studies have been conducted by domestic and foreign scholars, and most of them are conducted from the perspectives of public investment and fiscal policy. From the perspective of public investment, existing studies have found that government public investment helps to reduce production costs and increase productivity [31,32]. From the perspective of fiscal policy, different scholars possess different views. First, the study argues that fiscal decentralization helps improve the efficiency of local government resource allocation and GDP growth rate [33], Active fiscal policy can effectively improve the quality and efficiency of economic growth [34–37]. Second, the study concludes that there are different effects of different aspects of fiscal spending on high-quality economic development [38–40]. Third, the study concluded that regional imbalance and sectoral imbalance in government investment, excessive investment in for-profit projects crowding out space for private enterprises, and sloppy management of investment projects would weaken the efficiency of government investment [41].

In summary, different scholars have different understandings of economic quality development and consider constructing indicators of economic quality development from different levels to study the impact of different factors on economic quality development. On the one hand, government investment can effectively improve production efficiency, promote rational allocation of resources, reduce pollution emissions, and promote high-quality economic

development; on the other hand, government investment can also lead to regional and sectoral development imbalances and other problems, and produce a "crowding-out effect" on foreign investment and the private economy, hindering high-quality economic development. It is undeniable that scholars have made important contributions to the study of high-quality economic development, but no scholars have yet studied the impact of government investment behavior on high-quality economic development from the perspective of three industries based on the "Five Development Concepts". Based on the existing research, this paper selects indicators from five dimensions: innovation, coordination, greenness, openness and sharing, uses principal component analysis to construct a system of indicators for high-quality economic development, and uses the spatial Durbin model to study the impact of government investment behavior in the three industries on high-quality regional economic development and its sub-indicators.

## 3. Theoretical mechanism and research hypothesis

### 3.1 Primary industry government investment and high quality economic development

The primary industry is the basic industry of our national economy and has an important position in our national economic system. The impact of primary government investment on high-quality economic development is mainly manifested in two forces of positive promotion and negative inhibition. First, the government's investment in the primary industry helps promote the positive force of high-quality economic development. A large part of government investment in the primary industry is used for agricultural production expenditure, including investment in agricultural production and subsidies for agricultural producers. Agricultural production expenditure can effectively improve the technical level of agricultural production, promote large-scale agricultural production, improve agricultural production efficiency, enhance the production enthusiasm of agricultural producers, improve farmers' income and consumption levels, and promote high-quality economic development [42,43]. Rural infrastructure construction expenditure is also an important part of government investment in the primary sector. Rural infrastructure is a public good with strong externalities, and rural residents are usually unwilling and unable to invest in it, so the government usually invests in its construction, and these investments can effectively enhance the rural production and living environment and promote high-quality rural economic development. Secondly, the government's investment in the primary industry is not conducive to the promotion of high-quality economic development of negative inhibitory forces. Although the government's investment in agriculture can effectively enhance the scale of agricultural production, but in the process of modern agricultural production, the use of pesticides, fertilizers and other chemical products will also increase significantly, which has a negative impact on the production and living environment in rural areas. In addition, compared with the secondary and tertiary industries, the production efficiency of the primary industry is relatively low, increasing investment in the primary industry will have a crowding-out effect on the secondary and tertiary industries, which is not conducive to the overall high-quality economic development. Overall, the role of government investment in the primary industry on high-quality economic development depends on the magnitude of the force of these two aspects. Considering the actual situation in China, the use of chemical fertilizers and pesticides in agricultural production is quite common, and agricultural pollution is more serious. At the same time, China's small farmer ideology is difficult to reverse in the short term, and it is difficult to achieve rapid scale production and rapid improvement of technology level in general. Therefore, the negative inhibitory effect

of government investment in the primary industry on high-quality economic development is greater than the positive promoting effect.

**Hypothesis 1.** Government investment in the primary sector is not conducive to the improvement of the level of quality economic development.

## 3.2 Secondary industry government investment and high quality economic development

The secondary sector is the ballast of China's economic development and plays an important role in driving China's economic development. The impact of government investment in the secondary sector on the quality development of the economy also has two forces. On the one hand, government investment in the secondary industry has a positive role in promoting high-quality economic development. In recent years, China's economic development has entered a new normal, economic growth has slowed down, the industrial structure has been optimized, and the driving factors have shifted from factor-driven and investment-driven to innovation-driven. In this context, the Chinese government has paid more attention to the quality of economic development, increased investment in high-tech, green and other industries, and the level of technology and production efficiency has continued to improve, and the output value of high-tech, green and other industries has increased significantly, promoting high-quality economic development. On the other hand, government investment in the secondary industry has a negative inhibiting effect on high-quality economic development. China's secondary industry accounts for a large share of the overall economy, and problems such as high energy consumption, high emissions and low efficiency persist. Industry is a major contributor to China's energy consumption, carbon emissions and pollution emissions. In 2020, carbon emissions from five major industries, including electricity, iron and steel, cement, petrochemicals and coal chemistry, account for about 73% of national emissions, and government investment in these basic industries accounts for a large share [44]. With China's economic and social development, people's demand for material and energy is increasing, and the government's increased investment in power, steel and other industries will increase energy consumption and carbon and pollutant emissions, which will have a negative impact on high-quality economic development. Overall, the direction of operation of government investment in the secondary sector for high-quality economic development depends on the magnitude of the forces acting in both directions. Considering the current situation of China's economic development, in general, the current development of high-tech, green and environmental industries is relatively weak, while the proportion of electric power, steel and other industries is relatively large, government investment in the secondary industry has a greater negative inhibiting effect on the high quality of the economy.

**Hypothesis 2.** There is a negative inhibitory effect of government investment in the secondary sector on high-quality economic development.

## 3.3 Tertiary industry government investment and high quality economic development

The level of development of the tertiary sector is an important indicator of the degree of socialization of production and the level of development of the market economy. The development of tertiary industry can effectively promote the development of market economy, optimize the allocation of social resources (including natural resources, capital and labor), and is also an important way to improve the overall effectiveness and efficiency of the national economy. The tertiary sector, also known as the service sector, includes industries such as finance, tourism, transportation, and business services. Overall, these industries have relatively low energy

consumption and pollution emissions, low entry barriers, small investments, relatively high industrial added value, which can effectively raise people's income, and government investment in the tertiary sector can positively contribute to high-quality economic development.

**Hypothesis 3.** Government investment in the tertiary sector can positively contribute to high-quality economic development.

## 4. Variable selection and model construction

This section may be divided by subheadings. It should provide a concise and precise description of the experimental results, their interpretation, as well as the experimental conclusions that can be drawn.

### 4.1 Variable selection

**4.1.1 Explained variable.** With the help of SPSS software and principal component analysis, the article calculates the high-quality economic development indicators of various provinces and cities in China that constructs an indicator system of high-quality economic development from five aspects of innovation, coordination, greenness, openness and sharing, and carries out empirical analysis as explained variables. Table 1 shows the constructed economic high-quality development indicator system, which includes 5 first-level indicators, 18 second-level indicators, and 30 third-level indicators. The indicators to measure high-quality economic development include positive indicators, reverse indicators and moderate indicators. In order to explain high-quality economic development better, the article adopts the form of positive indicators, the form of reciprocal of reverse indicators and the form of reciprocal of absolute value of deviation for moderate indicators.

**4.1.2 Explanatory variables.** Based on the availability of data and existing research, the article selects indicators of government investment from the perspective of three major industries to analyze the impact on high-quality economic development (as shown in Table 2).

**Government investment in the primary industry.** Comprehensive agricultural development is an activity in which the government sets up special funds for the comprehensive development and utilization of agricultural resources in order to protect and support agricultural development, improve comprehensive agricultural production capacity and comprehensive benefits. Agricultural comprehensive development is an important manifestation of the government's support for the development of the primary industry. The article selects the proportion of the investment in agricultural comprehensive development projects to the fiscal expenditure as an indicator to measure the government's investment in the primary industry.

**Table 1. Indicator system of high-quality economic development.**

| First-Class Indicators | Second-Class Indicators | Third-Class Indicators | Index Interpretation | Indicator properties |
|---|---|---|---|---|
| Innovation | Innovation investment | The capital of R&D investment | R&D internal expenditure divided by total GDP | Positive |
| | | The time of R&D investment | R&D full-time staff equivalent divided by total population | Positive |
| | | The personnel of R&D investment | The number of employees in R&D institutions divided by the total population | Positive |
| | innovation capability | The growth rate of total factor productivity | DEA Malmquist index [1] | Positive |
| | | Labor productivity | Total GDP divided by total employed population | Positive |
| | | Capital productivity | Total GDP divided by capital stock | Positive |
| | Innovative achievements | Number of patents granted per 10,000 people | Patent grants divided by total population | Positive |

(*Continued*)

**Table 1.** (Continued)

| First-Class Indicators | Second-Class Indicators | Third-Class Indicators | Index Interpretation | Indicator properties |
|---|---|---|---|---|
| Coordination | Harmonious development between cities and countrysides | Income gap between urban and rural areas | The per capita disposable income of urban residents divided by the per capita disposable income of rural residents | Reverse |
| | | Urbanization rate | Urban population divided by total population | Positive |
| | Regional coordination | Regional economic disparity | Regional per capita GDP divided by national per capita GDP [2] | Reverse |
| | Industrial coordination | Industrial structure | The added value of the tertiary industry divided by the added value of the secondary industry | Positive |
| | Investment and consumption coordination | Investment consumption ratio | Investment divided by consumption | Moderate |
| Greenness | Pollution emission | Industrial wastewater discharge intensity | Industrial wastewater discharge intensity | Reverse |
| | | Industrial Sulfur Dioxide (SO$_2$) Emission Intensity | Industrial sulfur dioxide (SO2) emissions divided by total GDP | Reverse |
| | | Industrial smoke (powder) layer emission intensity | Industrial smoke (powder) layer emissions divided by total GDP | Reverse |
| | | Industrial solid waste generation intensity | The amount of industrial solid waste generated divided by the total GDP | Reverse |
| | Energy consumption | Energy consumption per ten thousand yuan of GDP | Total energy consumption divided by total GDP | Reverse |
| | | Electricity consumption per 10,000 yuan GDP | Electricity consumption divided by total GDP | Reverse |
| Openness | Openness to trade | Proportion of total import and export in GDP | Total imports and exports divided by total GDP | Positive |
| | Openness for investment | Proportion of actual foreign direct investment in GDP | Actual foreign direct investment divided by total GDP | Positive |
| | | Proportion of total industrial assets of foreign businessmen and Hong Kong, Macao and Taiwan businessmen | The proportion of the total assets of industrial enterprises above designated size divided by the total assets of industrial enterprises above designated size | Positive |
| | Tourism openness | International tourism foreign exchange earnings as a percentage of GDP | Foreign exchange income of international tourism divided by GDP | Positive |
| | | Number of inbound overnight tourists per 100 people | Number of inbound overnight visitors divided by total population | Positive |
| Sharing | Sharing of social security | Participation rate of basic old-age insurance for urban workers | The number of participants in the basic endowment insurance for urban workers is divided by the urban population | Positive |
| | Sharing of housing | The actual sales area of urban per capita residential housing | Actual sales area of urban residential housing divided by urban population | Positive |
| | Traffic sharing | Road mileage per capita | Total road mileage divided by total population | Positive |
| | Educational sharing | Years of education per capita | Calculated by multiplying the number of years at each educational level by its proportion in the total population [3] | Positive |
| | Employment sharing | Urban registered unemployment rate | Urban registered unemployment rate | Reverse |
| | Medical sharing | Number of health technicians per 10,000 people | Number of health technicians divided by total population | Positive |
| | | Number of beds in health institutions per 10,000 people | Number of beds in health institutions per 10,000 people | Positive |

Note: 1. The article uses GDP as the output indicator, and capital and labor as the input indicators to calculate the total factor productivity. Among them, the capital is represented by capital stock, and the capital stock is calculated according to the research method of Zhang Jun et al. (2004) [45]; the labor is represented by the total employed population.

2. Since the overall economic development level of the country cannot be compared with other regions, the national indicator system does not include the indicator of regional economic disparity when measuring coordination.

3. It is calculated by multiplying the number of years of each educational level by its proportion of the total population, which is set as 0 years for illiteracy, 6 years for primary school, 9 years for junior high school, 12 years for high school and secondary school, and 16 years for junior college and above.

**Table 2. Meaning of each variable.**

| Variable types | Variable name | Index | unit | Symbol |
|---|---|---|---|---|
| Explained variable | The level of high-quality economic development(*economic*) | Innovation | - | *innovative* |
| | | Coordination | - | *coordination* |
| | | Greenness | - | *sustainability* |
| | | Openness | - | *openness* |
| | | Sharing | - | *sharing* |
| Explanatory variables | Government investment in the primary industry | Agricultural comprehensive development project input divided by fiscal expenditure | % | *primary* |
| | Government investment in the secondary industry | The total assets of state-owned and state-controlled industrial and construction enterprises divided by the total assets of industrial and construction enterprises above designated size | % | *secondary* |
| | Government investment in the tertiary industry | State-owned tertiary industry fixed asset investment divided by the total fixed asset investment of the whole society | % | *tertiary* |
| Control variable | Industrial structure | The added value of the tertiary industry divided by GDP | % | *industry* |
| | Environmental governance | Total investment in industrial pollution control divided by GDP | % | *pollution* |
| | The size of population | Population density | per square kilometer | *population* |
| | Population quality | Population over the age of 6 divided by the population over the age of 6 | % | *education* |
| | Financial development | The balance of RMB deposits and loans of financial institutions divided by GDP | % | *financial* |

**Government investment in the secondary industry.** The capital of state-owned and state-holding enterprises is wholly or mainly invested by the state, and all or most of its assets are owned by the state. The government has ownership or control over state-owned enterprises, and the will and interests of the government determine the behavior of state-owned and state-controlled enterprises. Drawing on the research experience of Yan Lei (2005), the article selects the proportion of the total assets of state-owned and state-controlled industrial and construction enterprises to the total assets of industrial and construction enterprises above designated size as an indicator to measure government investment in the secondary industry [46].

**Government investment in the tertiary industry.** The government is the main investor in state-owned enterprises and the owner of state-owned enterprise assets. The investment behavior of state-owned enterprises can generally represent the will of the government. Based on the research experience of Wang Danli (2014) [47], the proportion of the fixed asset investment of the state-owned tertiary industry in the total fixed asset investment of the whole society is selected as the index to measure the government investment of the tertiary industry [47].

**4.1.3 Control variable.** Based on the availability of data and on the basis of existing studies, this paper selects indicators of government investment behavior from the perspective of the three major industries to analyze the impact on high-quality economic development (as shown in Table 2).

**Industrial structure.** Industrial structure has an important impact on the quality of economic development. In general, the optimization of industrial structure can effectively improve the efficiency of economic development, reduce energy consumption and pollution emission, thus improving the quality of economic development. In this paper, the ratio of the added value of tertiary industry to the added value of secondary industry is selected as an indicator to measure the industrial structure.

**Environmental governance.** Environmental pollution is an important factor affecting the quality of economic development, and good environmental governance capacity can effectively improve the quality of economic development. In this paper, the completed investment

amount of industrial pollution control as a proportion of industrial value added is selected as an indicator of environmental governance.

**Population size.** Generally speaking, areas with higher population density are more crowded and have relatively larger domestic pollution emissions, thus not conducive to the improvement of economic development quality. In this paper, population density is selected as an indicator to measure the population size.

**Population quality.** The quality of the population has a certain influence on the quality of economic development. Areas with high population quality have more harmonious and stable societies and higher productivity, thus improving the quality of economic development. In this paper, the proportion of the population with tertiary education or above to the population above 6 years old is selected as an indicator of population quality.

**Level of financial development.** Generally speaking, regions with higher level of financial development are more efficient in corporate investment and financing, and enterprises can obtain better development, thus improving the quality of economic development. In this paper, the proportion of RMB deposit and loan balance of financial institutions to GDP is selected as an indicator to measure the level of financial development.

## 4.2 Model building

The theoretical basis of the traditional econometric model is based on the independence of observed values, while the economic behavior between regions generally has spatial interaction or spatial effect (spatial correlation and spatial heterogeneity) [48]. If the influence of spatial effect is ignored, the model estimation would be biased and lack of explanation. In the spatial econometric model, it is only considered the spatial correlation of the explained variables by using the spatial lag model (SLM) and spatial error model (SEM), while the spatial Dubin model (SDM) considers not only the spatial correlation of the explained variables, but also the spatial correlation of the explained variables, which is more in line with the actual situation and can more effectively reflect the influence of the explained variables on the explained variables. Therefore, the article constructs a spatial Dubin model for analysis. Some scholars consider the superiority of the spatial Durbin model and construct the spatial Durbin model to study the economic quality development. Wang et al. (2021) studied the impact of environmental pollution and green finance on high-quality energy development in the Yangtze River Economic Zone by constructing a spatial Durbin model [49], Wang et al. (2021) studied the impact of green technological innovation on high-quality economic development by constructing a spatial Durbin model [50,51] used a spatial Durbin model to study the impact of environmental regulation on green total factor productivity [51]. Therefore, this paper also considers these advantages and draws on the research experience of scholars to construct a spatial Durbin model for analysis.

$$
\begin{aligned}
y_{it} = \rho W y_{it} + \alpha_0 + \alpha_1 lprimary_{it} + \alpha_2 lsecondary_{it} + \alpha_3 ltertiary_{it} + \sum_{m=1}^{n} \beta_m X_m \\
+ W\delta_1 lprimary_{it} + W\delta_2 lsecondary_{it} + W\delta_3 ltertiary_{it} + \sum_{m=4}^{n} W\delta_m X_m + \mu_{it}
\end{aligned}
\tag{1}
$$

Where, y represents the high-quality economic development and its sub-indices; primary represents the government investment in the primary industry; secondary represents the government investment in the secondary industry; tertiary represents the government investment in the tertiary industry; the "$l$" in front of a variable means taking the log of the variable; $X_m$ represents the control variable, including industrial structure, environmental governance,

**Table 3. Global Moran's I index of high-quality economic development from 2006 to 2016.**

| year | 2006 | 2007 | 2008 | 2009 | 2010 | 2011 | 2012 | 2013 | 2014 | 2015 | 2016 |
|---|---|---|---|---|---|---|---|---|---|---|---|
| Moran's I | 0.174 | 0.192 | 0.213 | 0.222 | 0.252 | 0.296 | 0.332 | 0.340 | 0.342 | 0.368 | 0.359 |
| Z value | 2.733 | 2.710 | 2.944 | 2.812 | 3.125 | 3.427 | 3.562 | 3.623 | 3.894 | 3.917 | 3.903 |
| P-value | 0.015 | 0.010 | 0.008 | 0.009 | 0.006 | 0.004 | 0.002 | 0.003 | 0.001 | 0.001 | 0.001 |

population size, population quality, financial development; $\alpha_k$, $\beta_k$ indicates the parameter to be estimated; $\mu_{it}$ represents the random error; $\rho$ is the spatial autocorrelation coefficient; $W$ is a spatial weight matrix constructed based on the adjacency principle, that is, geographically adjacent areas are assigned 1, and non-adjacent areas are assigned 0.

Due to the different statistical caliber of Hong Kong, Macao, Taiwan and other places, there are many missing data in Tibet Autonomous Region. The research scope of the article covers 30 provinces, municipalities and autonomous regions in Chinese Mainland (except Tibet), and the time dimension is 2006–2016. The original data in the article are all from China Statistical Yearbook, China Industrial statistical yearbook, China Financial Yearbook, China Fixed Assets Statistical Yearbook, China Tertiary Industry Statistical Yearbook, Compilation of Statistical Data in the 60 Years of New China, statistical yearbooks of provinces and cities over the years. Some missing data were estimated by interpolation, regression analysis and other methods. In order to eliminate the influence of price factors, economic indicators such as total GDP and per capita GDP are the data converted from the resident price index. For reducing the difference in the value of the value and increasing the stability of the data, the article performs logarithmic processing on the explanatory variables and the control variables.

### 4.3 Spatial autocorrelation analysis

With the help of GEODA software, the adjacency principle was used to construct the space weight matrix, and the global Moran's I index of high-quality economic development in China from 2006 to 2016 was calculated. It can be seen from Table 3 that the global Moran's I index of China's high-quality development of China's economy was positive and passed 5% of the significant test from 2006 to 2016. It shows that the level of China's high-quality economic development has a positive space correlation, and the space correlation is generally rising. That is because without considering other factors, there is a certain space overflow effect in the high-quality economic development of the region, such as the upgrading of the industrial structure in the region, which would drive the development of related industries in adjacent areas to a certain extent.

On the basis of global autocorrelation analysis, Moran's I scatter diagram can be used to further analyze the local spatial correlation of high-quality economic development. because of space limitations, only Moran's I scatter diagram of high-quality economic development in 2006 and 2016 is given here (as shown in Fig 1). It can be seen from Fig 1 that in 2006 and 2016, most of the high-quality economic development concentrated in the first and third quadrants, showing the characteristics of high and low agglomeration. It shows that without considering the impact of other factors, the high-quality economic development of 2006 and 2016 shows a significant space correlation.

## 5. An empirical analysis of the impact of government investment on high-quality economic development

In order to judge whether it is suitable to build a spatial econometric model for estimation, it is necessary to perform model testing at first. The results of the LM test show that LM (lag) and

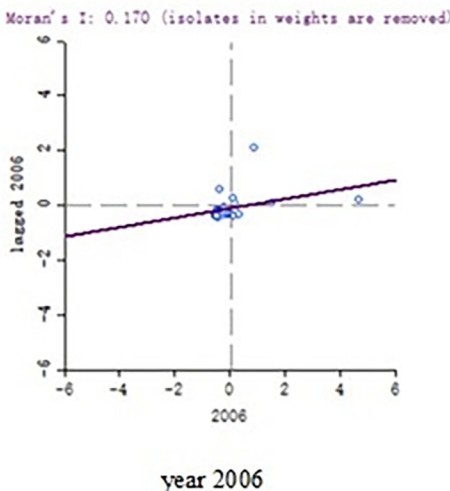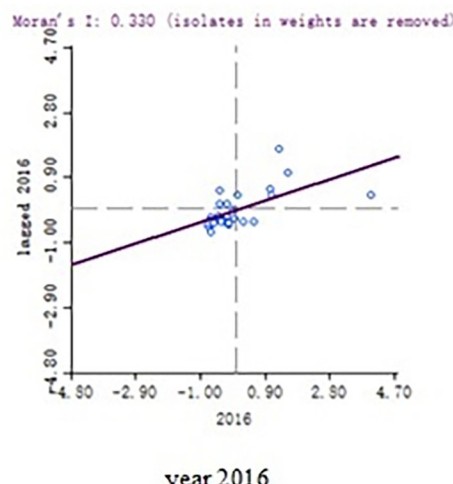

**Fig 1. Moran's I scatter diagram of spatial correlation of China's high-quality economic development in 2006 and 2016.**

LM (error) and robust LM (lag) and robust LM (error) of each model contain options that have passed the 10% significance test, indicating that it is more appropriate to build a spatial metrology model (as shown in Table 4). The results of the Hausman test show that the spatial fixed-effects model is more efficient than the spatial random-effects model. The LR test results show that both LR-test (SFE) and LR-test (TFE) have passed the 1% significance test, indicating that the space-time two-dimensional fixed effects are more efficient. All of the results of Wald test and LR test in Table 5, Wald spatial lag, LR spatial lag, Wald spatial error and LR spatial error have passed the significance test of 1%, indicating that spatial Doberman model (SDM) cannot be simplified into spatial lag model (SLM) and spatial error model (SEM). To sum up, SDM with two-dimensional fixed effects in space and time should be selected for estimation.

The estimation results of the SDM of the two-dimensional fixed effects of space and time in Table 5 show that the spatial autocorrelation coefficient of high-quality economic development is negative. It shows that there is a negative spatial spillover effect in high-quality economic development, and the improvement of the level of high-quality economic development in the region would have an inhibitory effect on the high-quality economic development of adjacent regions. The high-quality economic development between adjacent regions should have a positive spatial correlation, but under the influence of government forces and other factors, all

**Table 4. LM, LR and Hausman test of high-quality economic development model.**

| Inspection type | OLS | Spatial fixation effect | Time fixed effect | Space-time two-dimensional fixed effect |
|---|---|---|---|---|
| LM(lag) | 0.530 | 38.303*** | 12.800*** | 8.882*** |
| Robust LM(lag) | 3.431* | 64.305*** | 10.116*** | 29.837*** |
| LM(error) | 34.291*** | 3.151* | 2.794* | 0.975 |
| Robust LM(error) | 37.192*** | 29.153*** | 0.110 | 21.930*** |
| LR-test(SFE) | 557.083*** | | | |
| LR-test(TFE) | 110.378*** | | | |
| Hausman | -103.461*** | | | |

Note: The symbols ***

** and * indicate that the variables are significant at the 1%, 5% and 10% significance levels, respectively.

**Table 5. Estimated results of the SDM for high-quality economic development.**

| Variable | OLS | Spatial fixation effect | Time fixed effect | Space-time two-dimensional fixed effect |
|---|---|---|---|---|
| *lprimary* | -0.121*** | -0.118*** | -0.247*** | -0.0841** |
| | (-2.866) | (-3.081) | (-6.493) | (-2.285) |
| *lsecondary* | -0.151*** | -0.0864 | -0.366*** | -0.125* |
| | (-3.097) | (-1.104) | (-7.668) | (-1.669) |
| *ltertiary* | -0.0438 | 0.0733 | 0.0222 | 0.0990** |
| | (-0.647) | (1.522) | (0.318) | (2.174) |
| rho | | 0.0801 | -0.014 | -0.097 |
| | | (1.292) | (-0.203) | (-1.470) |
| Control variable | Yes | Yes | Yes | Yes |
| R-squared | 0.861 | 0.985 | 0.926 | 0.987 |
| Log-likelihood | | 296.829 | 37.305 | 322.218 |
| Wald_spatial_lag | | 110.180 | 73.166 | 69.526 |
| | | [0.000] | [0.000] | [0.000] |
| LR_spatial_lag | | 100.569 | 64.945 | 63.744 |
| | | [0.000] | [0.000] | [0.000] |
| Wald_spatial_error | | 115.836 | 77.906 | 66.116 |
| | | [0.000] | [0.000] | [0.000] |
| LR_spatial_error | | 121.856 | 72.562 | 61.049 |
| | | [0.000] | [0.000] | [0.000] |
| Observations | 330 | 330 | 330 | 330 |

Note: The symbols * * *, * * and * respectively indicate that the variables are significant at the significance level of 1%, 5% and 10%. The numbers in parentheses of OLS are t statistics, the numbers in parentheses of SDM are Z statistics, and the numbers in parentheses are p-value values.

regions are competing to take measures to promote the high-quality economic development of their own regions. There is a certain competitive relationship between adjacent regions in terms of resources, talents, capital and market, which leads to a certain inhibitory effect on the economic development between adjacent regions. The article analyzes the impact of government investment on high-quality economic development from the perspective of direct effect and indirect effect (as shown in Table 6).

The direct effect of government investment in the primary industry on high-quality economic development is negative, and it has passed the 5% significance test, indicating that

**Table 6. Spatial effect decomposition table of SDM under space-time two-dimensional fixed effects of high-quality economic development.**

| Variable | Direct effect | Indirect effect | Total effect |
|---|---|---|---|
| *lprimary* | -0.086** | -0.029 | -0.115** |
| | (-2.291) | (-0.543) | (-2.071) |
| *lsecondary* | -0.110 | -0.573*** | -0.684*** |
| | (-1.407) | (-3.566) | (-4.091) |
| *ltertiary* | 0.102** | -0.177** | -0.075 |
| | (2.225) | (-2.117) | (-0.948) |

Note: The symbols ***

** and * indicate that the variables are significant at the 1%, 5% and 10% significance levels, respectively, and the z-statistics are in parentheses.

government investment in the primary industry has a negative impact on the high-quality economic development of the regions. There may be three reasons: first, government investment in the primary industry would increase the use of pesticides and fertilizers that may be harmful to plants and water.; second, it is relatively low on the production efficiency of the primary industry and the added value of products, and excessive government investment in the primary industry is not conducive to industrial transformation and upgrading; third, at present, China is still in the primary stage of socialist development, and the rural population still accounts for about 40% of China's total population, then increasing the government's investment in the primary industry would have a crowding out effect on the rural population and reduce its income, which is not conducive to the stability of social development. The indirect effect is also negative, but not significant, indicating that government investment in the primary industry has a limited impact on the high-quality economic development of adjacent regions. The main reason is that the pesticides and fertilizers used in large quantities in agricultural production will follow the water flow to the adjacent areas, which will have a negative impact on the ecological environment of the adjacent areas.

The direct effect of government investment in the secondary industry on high-quality economic development is negative, but it fails to pass the 10% significance test, indicating that government investment in the secondary industry has a negative impact on high-quality economic development in the region, but the impact is not significant. Due to historical and industrial characteristics, state-owned and state holding enterprises account for a large proportion of industrial enterprises with high pollution and high energy consumption. Further increasing the government's investment in the secondary industry would bring pressure on the ecological environment to a certain extent, which is not conducive to the transformation and upgrading of industrial structure and the improvement of production efficiency, and have an adverse impact on the high-quality development of economy. The indirect effect is also negative and passed the 1% significance test, indicating that the government investment in the secondary industry has a significant negative spatial spillover effect, which has a significant adverse impact on the high-quality economic development of adjacent regions. This is mainly because the pollutants generated by industrial development in the region flow to adjacent areas through water and air, which adversely affects the too environment of adjacent areas.

The direct effect of government investment in the tertiary industry on high-quality economic development is positive, and it has passed the 5% significance test, indicating that government investment in the tertiary industry has a significant positive impact on high-quality economic development. The tertiary industry has some advantages, such as small investment, fast absorption, good benefits and large employment capacity. The government's investment in the tertiary industry can promote the development of the tertiary industry, enhance the confidence of private enterprises in investing in the tertiary industry and promote the high-quality development of the economy in the region. The indirect effect is negative and has passed the 5% significance test, indicating that there is a significant negative spatial spillover effect in the government investment in the tertiary industry. The main reason is that there is a certain competitive relationship and siphoning effect between the region and the neighboring regions in the development of tertiary industry, and the development of tertiary industry in the region will have a certain inhibiting effect on the neighboring regions.

## 6. Empirical test of the role of government investment on sub-indicators of high-quality economic development

Similarly, the SDM under the two-dimensional fixed effect of space and time is selected to analyze the impact of government investment on sub indicators of high-quality economic

**Table 7. SDM estimation results of economic development quality sub-indices.**

| Variable | Innovation | Coordination | Greenness | Openness | Sharing |
|---|---|---|---|---|---|
| *lprimary* | -0.0353 | 0.0343 | -0.388*** | -0.0154 | -0.288*** |
| | (-0.579) | (1.079) | (-3.338) | (-0.259) | (-4.113) |
| *lsecondary* | 0.306** | -0.0834 | 0.0560 | -0.828*** | -0.448*** |
| | (2.521) | (-1.291) | (0.238) | (-6.849) | (-3.156) |
| *ltertiary* | 0.181** | 0.0736* | -0.383*** | 0.0903 | 0.247*** |
| | (2.432) | (1.889) | (-2.645) | (1.221) | (2.882) |
| rho | 0.392*** | 0.0759 | -0.293*** | 0.0339 | 0.341*** |
| | (7.404) | (1.086) | (-5.413) | (0.466) | (5.290) |
| Control variable | Yes | Yes | Yes | Yes | Yes |
| R-squared | 0.978 | 0.994 | 0.917 | 0.978 | 0.970 |
| Log-likelihood | 154.435 | 370.685 | -60.251 | 165.254 | 105.885 |
| Observations | 330 | 330 | 330 | 330 | 330 |

Note: The symbols ***

** and * indicate that the variables are significant at the 1%, 5% and 10% significance levels, respectively, and the z-statistics are in parentheses.

development. As can be seen from Table 7, the spatial autocorrelation coefficients of innovation, coordination, openness and sharing models are positive, while that of greenness model is negative. The spatial effect of the sub-indicators of high-quality economic development is also decomposed, and the impact of government investment on high-quality economic development is analyzed from the perspective of direct effect and indirect effect (as shown in Table 8).

## Innovation

The direct effect of government investment in the primary industry on innovation is negative, but it fails to pass the 10% significance test, but it is positive in the secondary industry and tertiary industry, and the effect in the tertiary industry has passed the 10% significance test, while the secondary industry is not significant. It shows that the government's investment in the

**Table 8. Decomposition table of spatial effect of SDM under two-dimensional fixed effects of space and time for sub-indices of high-quality economic development.**

| Effect | Variable | Innovation | Coordination | Greenness | Openness | Sharing |
|---|---|---|---|---|---|---|
| Direct effects | *lprimary* | -0.070 | 0.028 | -0.389*** | -0.025 | -0.317*** |
| | | (-1.090) | (0.843) | (-3.237) | (-0.433) | (-4.361) |
| | *lsecondary* | 0.208 | -0.086 | 0.087 | -0.835*** | -0.529*** |
| | | (1.612) | (-1.362) | (0.362) | (-6.851) | (-3.526) |
| | *ltertiary* | 0.136* | 0.077* | -0.274* | 0.095 | 0.251*** |
| | | (1.895) | (1.921) | (-1.773) | (1.357) | (2.922) |
| Indirect effect | *lprimary* | -0.491*** | -0.227*** | 0.069 | 0.057 | -0.311** |
| | | (-3.558) | (-4.427) | (0.452) | (0.622) | (-2.155) |
| | *lsecondary* | -1.366*** | -0.230 | -0.178 | -1.065*** | -1.238** |
| | | (-3.208) | (-1.562) | (-0.396) | (-3.899) | (-2.671) |
| | *ltertiary* | -0.572** | 0.170** | -1.243*** | 0.721*** | 0.093 |
| | | (-2.714) | (2.125) | (-5.196) | (5.115) | (0.417) |

Note: The symbols ***

** and * indicate that the variables are significant at the 1%, 5% and 10% significance levels, respectively, and the z-statistics are in parentheses.

primary industry has an adverse effect on the innovation of the regional economic development, but it is not significant, while the investment in the secondary and tertiary industries has a promoting effect, of which the tertiary industry is more significant. On the one hand, the primary industry is a traditional industry with small innovation space and relatively small benefits of innovation investment. On the other hand, affected by China's history and culture, topography and population, most of the government investment in the primary industry is to give farmers production subsidies and expand production scale, while the investment in technological innovation is relatively small, so the impact on innovation is limited. While, there are good prospects for the development of innovation in the secondary and tertiary industries, that the investment of innovation can achieve greater benefits for the economic and society, and increasing investment in the secondary and tertiary industries is conducive to innovation. The indirect effects of government investment on innovation in the three industries are all significantly negative, indicating that government investment behaviors have significant negative spatial spillover effects on the innovation and development of adjacent regions.

## Coordination

The direct effect of the government investment of the primary industry and the tertiary industry on the coordination is positive, while it is negative for the investment in secondary industry. Among them, the direct effect of tertiary industry has passed the 10% significance test, while the effect of the primary and secondary industries is not significant. It shows that the government's investment in the primary and tertiary industries has a positive impact on the coordination of the region. The impact of the investment in the tertiary industry is significant, while the investment in the secondary industry has a negative impact, but not significant. Generally speaking, most of the employees in the primary industry are groups with relatively lower level of income. The increasing of government's investment in the primary industry can improve the income of farmers to a certain extent, narrow the income gap between urban and rural areas, and promote the coordination of urban and rural development. Most of the employees in the secondary industry live in urban areas with relatively higher levels of income. Then the increasing of government's investment in the secondary industry would widen the income gap between urban and rural areas. In the context of the new normal, the increasing of the government's investment in high-tech industries and increasing the introduction and training of high-level talents would further expand the income gap, which is not conducive to the coordinated development of economy. The tertiary industry has the characteristics of high employment capacity. More jobs would be provided, the living standards would be improved, and the coordination of economic development would be enhanced by increasing the government's investment in the tertiary industry. The indirect effect of government investment in primary industry and secondary industry on coordination is negative, in which the indirect effect of the primary industry is more significant, and the indirect effect of the government investment of the tertiary industry is significantly positive. It shows that the government's investment in the primary industry can have a significant negative spatial spillover effect on the coordination of economic development in adjacent areas, but it is not significant that the investment in the secondary industry has a negative spatial spillover effect, and spatial spillover effect of the investment in the tertiary industry is significantly positive.

## Greenness

The direct effect of government investment in the primary industry and the tertiary industry on greenness is significantly negative, while the direct effect of government investment in the secondary industry on greenness is positive that it is not significant. The level of greenness

would be reduced that the government increases investment and expands the production scale in the primary industry. Because it would increase the use of pesticides, fertilizers and other chemical substances, and the water and vegetation would be damaged. In the context of the new normal of the economy, the government's investment in the secondary industry pays more attention to the development of innovative industries, so that the level of greenness would be improved. The tertiary industry is in a stage of rapid development with a relatively low level of development. In order to enhance the level of the tertiary industry, local governments scramble to seize the development opportunities of the tertiary industry, but lack the overall development plan, so that the development resources cannot be used effectively. Moreover, most of the government's investment in the tertiary industry is concentrated on engineering construction, which would consume a certain amount of resources, endanger the ecological environment, and reduce the level of greenness. The indirect effect of the government's investment in the primary industry on the impact of greenness is positive, but not significant, indicating that the positive spatial spillover effect is not significant. And the indirect effects of investment in the secondary and tertiary industries are negative, indicating that there is a negative spatial spillover effect, among which the investment behavior in the tertiary industry plays a more significant role.

## Openness

The direct effects of government investment in the primary industry and the secondary industry on openness are both negative. Among them, the role of government investment in the secondary industry has passed the 1% significance test. The direct effect of government investment in the tertiary industry is positive, but not significantly. It shows that the government's investment in the primary industry and the secondary industry can have an adverse impact on the expansion and opening-up of the region, of which the adverse impact of the secondary industry is more significant, and the investment in the tertiary industry can promote the expansion and opening-up of the region, but the promotion effect is not significant. China has a large population and is a large importer of agricultural products. The increasing in agricultural product output brought by the increasing of the government's investment in the primary industry would reduce China's imports of agricultural products, and even reduce the degree of openness. The government's investment in the secondary industry would have a certain crowding-out effect on foreign capital, which is not conducive to the improvement of openness. The government's investment industry would promote the development of the tertiary industry and improve the degree of opening to the outside world, that it would attract more foreign tourists to China, and increase foreign exchange income from tourism. The indirect effects of government investment in the primary and tertiary industries on the impact of openness are both positive, indicating that there is a positive spatial spillover effect, which is more significant in the tertiary industry. But the indirect effect of investment in the secondary industry is significantly negative, indicating that there is a significant negative spatial spillover effect.

## Sharing

The direct effects of government investment in the primary industry and the secondary industry on the sharing are significantly negative, and it is significantly positive in the tertiary industry. It shows that the government's investment in the primary and tertiary industries is not conducive to promoting the sharing of economic development results, while the tertiary industry is the opposite. Government investment in the tertiary industry can effectively promote the sharing of economic development gains. Although the government's investment in the

primary and secondary industries can enhance people's income to a certain extent, but it would occupy the time for rest and affect people's health, which is not conducive to the sharing of the achievement of economic development. But the tertiary industry is different, that government investment in the tertiary industry can not only promote the improvement of people's income, but also realize the sharing of the achievement of economic development that provide people with more places for leisure, entertainment and sightseeing to improve people's quality of life. The indirect effect of the government's investment in the primary and secondary industries on the sharing effect is significantly negative, indicating that there is a significant negative spatial spillover effect. The indirect effect of the government's investment in the tertiary industry on the sharing effect is positive, indicating that there is a positive spatial spillover effect, but it is not significant.

## 7. Conclusions and suggestions

The article selects indicators from the five dimensions of innovation, coordination, greenness, openness and sharing to build an evaluation system of high-quality economic development. Based on the panel data of 30 provinces, municipalities and autonomous regions (excluding Tibet) in Chinese Mainland, the time dimension is from 2006 to 2016. The principal component analysis is used to calculate the indicators of high-quality economic development. The spatial correlation is analyzed in the article. The indicators of government investment are selected from the perspective of three industries to analyze the direct and indirect effects of government investment on high-quality economic development and its sub indicators by building a spatial Dubin model. Conclusions: first, there is a significant positive spatial correlation between China's high quality economic development and the impact of high-quality economic development in the region on neighboring regions. Second, the government's investment behavior in different industries has different impacts on the high-quality economic development. The government's investment behavior in primary and secondary industries will increase the use and emission of pollutants, which is not conducive to the high-quality economic development of the region, while the investment behavior in tertiary industries has the advantages of high added value, low pollution, driving employment and increasing income, which can effectively promote the high-quality economic development of the region. Third, there are differences in the impact of government investment behavior in different industries on different aspects of high-quality economic development. The government's investment in the primary industry has an adverse impact on the greenness and sharing of economic development, the negative impact on innovation and openness is not significant, and the positive impact on coordination is not significant. The government's investment in the secondary industry has a significant adverse effect on the openness and sharing of economic development in the region, while the promotion effect on innovation and greenness, and the negative effect on coordination is not obvious. The government's investment in the tertiary industry has a significant role in promoting the innovation, coordination and sharing of local economic development, and has no significant role in promoting openness, but has a significant adverse effect on greenness. The government's investment in the tertiary industry has a significant role in promoting the innovation, coordination and sharing of local economic development, and has no significant role in promoting openness, but has a significant adverse effect on greenness. Fourth, government investment would not only have an effect on the high-quality economic development of the region, but also have a certain spatial spillover effect, which would have an impact on the high-quality economic development of adjacent regions.

According to the research conclusions, we can get the following enlightenment: First, improving agricultural production methods and enhancing agricultural production efficiency.

The government's investment in the primary industry should focus on improving agricultural production efficiency, improving production methods, promoting large-scale production, increasing investment in agricultural technology research and development, improving the technical level of agricultural production, increasing the added value of agricultural products, and developing green and ecological agriculture. Second, to promote industrial transformation and upgrading, to achieve innovative development. The government's investment in the secondary industry focuses on high-tech industries, increasing investment in science and technology innovation, supporting the development of innovative industries, and promoting the transformation and upgrading of industrial structure; increasing the training and introduction of talents, focusing on the hierarchy and diversity of talent training, to cultivate not only high-level research talents in China, but also to cultivate skilled talents who can adapt to various technical jobs, for the efficient development of China's economy to escort the efficient development of Chinese economy. Third, fully exploit the development advantages and strengthen the development of industrial integration. The government should increase investment in the tertiary industry, fully exploit the development advantages of the region, rationalize the planning layout, strengthen cooperation with non-state enterprises, and guide the integration of the primary and secondary industries with the tertiary industry. Fourth, strengthen regional cooperation to achieve common development. Different regions should strengthen cooperation and communication, establish cooperation mechanisms, enhance mutual understanding, realize complementary advantages, strengthen regional coordination, reduce zero-sum games in local economic exchanges, increase win-win results, and achieve common development.

The main contribution of this paper is to select indicators from five dimensions: innovation, coordination, greenness, openness, and sharing, to construct an evaluation system for high-quality economic development, to study the impact of government investment behavior on high-quality economic development and its different aspects from the perspective of three industries, and to find some variability in the government investment behavior in different industries on high-quality economic development and its different aspects of operations. Meanwhile, it is difficult to obtain more relevant data on government investment due to the limitation of limited data, while the time cut-off point is only up to 2016. In the future, we will further study more specific aspects based on this paper to investigate the impact of government investment behavior in specific industries on economic quality development.

## Supporting information

**S1 File. Basic data.** This file is mainly the base data used in this paper.
(PDF)

**S2 File. Description of data.** This file is mainly a description of the underlying data sources used in this paper.
(PDF)

## Author Contributions

**Conceptualization:** Ming Wang.

**Data curation:** Ming Wang.

**Formal analysis:** Ming Wang.

**Funding acquisition:** Weiming Liu.

**Methodology:** Ming Wang.

**Project administration:** Ming Wang.

**Supervision:** Weiming Liu.

**Writing – original draft:** Ming Wang.

**Writing – review & editing:** Ming Wang.

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
