## [Decision Letter · Decision Letter 0]

17 Nov 2022

PONE-D-22-25225An empirical analysis of the impact of Chinese government investment on high-quality economic development——A study based on Spatial Dubin ModelPLOS ONE

Dear Dr. Wang,

Thank you for submitting your manuscript to PLOS ONE. After careful consideration, we feel that it has merit but does not fully meet PLOS ONE’s publication criteria as it currently stands. Therefore, we invite you to submit a revised version of the manuscript that addresses the points raised during the review process.

The submission requires further revisions oriented mainly to prior literature discussion and quantitative framework.

We look forward to receiving your revised manuscript.

Kind regards,

Stefan Cristian Gherghina, PhD. Habil.

Academic Editor

PLOS ONE

Journal Requirements:

Reviewers' comments:

Reviewer's Responses to Questions

**Comments to the Author**

1. Is the manuscript technically sound, and do the data support the conclusions?

Reviewer #1: Partly

Reviewer #2: Yes

Reviewer #3: Yes

Reviewer #4: No

2. Has the statistical analysis been performed appropriately and rigorously? 

Reviewer #1: N/A

Reviewer #2: Yes

Reviewer #3: Yes

Reviewer #4: No

3. Have the authors made all data underlying the findings in their manuscript fully available?

Reviewer #1: Yes

Reviewer #2: Yes

Reviewer #3: Yes

Reviewer #4: No

4. Is the manuscript presented in an intelligible fashion and written in standard English?

Reviewer #1: No

Reviewer #2: Yes

Reviewer #3: Yes

Reviewer #4: No

5. Review Comments to the Author

Reviewer #1: Title: An empirical analysis of the impact of Chinese government investment on high-quality economic development——A study based on Spatial Dubin Model

Manuscript Number: PONE-D-22-25225

Journal: PLOS ONE

It is my pleasure to review the manuscript for the journal. In this work, the authors assessed the economic high-quality development index system using the principal component analysis, and then investigated the impact of impact of the government’s investment in the three industries on the high-quality development of regional economy by using the spatial Dubin model. The work presented is relevant to the Journal's field. The manuscript has got some potential. I would like to congratulate the author for a considerable amount of work that they have done. Especially, the authors uncovered that government investment in the secondary industry can enhance innovation, while the “crowding out effect” harms the openness and the sharing, discouraging high-quality economic development. This manuscript has provided a new case to better understanding of the impact of country risks, renewable energy, trade openness and urbanization on ecological sustainability. However, the manuscript needs further improved before to be accepted for publication. The reviewer has listed some specific comments that might be helpful of the author to further enhance the quality of the manuscript. Please consider the particular comments listed below.

Comment 1: Abstract. The abstract is well-structured and well-written. However, it should underscore the scientific value added of your paper in your abstract, rather than others.

Comment 2: section of Introduction. As pointed out in the reports of World Bank, UNEP and others, the COVID-19 pandemic has profoundly affected and changed the global and regional sustainability issues. Therefore, this section should not ignore the impact of the pandemic on carbon emission. Please consider citing following paper: https://doi.org/10.1016/j.scitotenv.2020.138915. Added these citations will certainly improve the practical significance of the research of this article.

Comment 3: section of literature review. The novelty of this paper should be further justified by highlighting main contributions to the existing literature. This could be clearly presented in the Literature review related work. Please consider please consider citing following papers e.g., (i) https://doi.org/10.1016/j.jclepro.2020.123838;(ii) https://doi.org/10.1016/j.spc.2021.02.031; (iii) https://doi.org/10.1016/j.scs.2021.103382; (iv) https://doi.org/10.1016/j.envres.2022.114575. There has already been a large number of literatures related to the panel data analysis and panel data model. There is a need to better elaborate the contribution of the work to the existing literature.

Comment 4: section of Variable selection and model construction The detailed information of Variable selection is impressive. However, it would be better to further highlight your improvement of the method and your innovation in methods.

Comment 5: sections of Empirical finding and Discussion. This section is well-structured. However, it would be better to discuss what your findings are different from the past works.

Comment 6: section of Conclusion and Policy Implication. Please make sure your conclusions' section underscore the scientific value added of your paper, and/or the applicability of your findings/results, as indicated previously. Basically, you should enhance your contributions, limitations, underscore the scientific value added of your paper, and/or the applicability of your findings/results and future study in this session.

Comment 7: There are still some occasional grammar errors through the revised manuscript especially the article ''the'', ''a'' and ''an'' is missing in many places, please make a spellchecking in addition to these minor issues. In addition, some sentences are too long to be easy to read. It is recommended to change to short sentences, which are easier to read.

Comment 8: References. Please check the references in the text and the list; You should update the reference.

Comment 9: line number is missing, please add it.

Reviewer #2: This study first establishes an index system for high-quality economic development, and examines the effect of government investment on high-quality economic development with Spatial Dubin Model.

I think this is an interesting topic. Also, spatial econometric method is an appropriate method to discuss the research question of this study. I have a few comments which may help the authors to improve their study.

1. In fact, high-quality economic development is a relatively new concept, and there are few studies to discuss its determinants, such as green finance, industrial co-agglomeration, and environmental regulation and so on. Thus, I suggest the authors to add some discussion in the literature review section. I would suggest following additions:

a) Yang, Y., Su, X., & Yao, S. (2021). Nexus between green finance, fintech, and high-quality economic development: Empirical evidence from China. Resources Policy, 74, 102445.

b) Zheng, H., & He, Y. (2022). How does industrial co-agglomeration affect high-quality economic development? Evidence from Chengdu-Chongqing Economic Circle in China. Journal of Cleaner Production, 371, 133485.

c) Wang, L., Wang, Z., & Ma, Y. (2022). Does environmental regulation promote the high-quality development of manufacturing? A quasi-natural experiment based on China's carbon emission trading pilot scheme. Socio-Economic Planning Sciences, 81, 101216.

2. Please add the theory or literature which can support for the selection of these control variables in Section 3.1.3. Furthermore, please further modify the grammar and expression of the MS.

3. It is necessary to add the robustness test of this MS. Meanwhile, please explain the possible reasons behind the empirical findings that the indirect effect of government investment in the primary industry on high-quality economic development is negative, but not significant, and the indirect effect of government investment in the secondary industry on high-quality economic development is significantly negative, and the indirect effect of government investment in the tertiary industry on high-quality economic development is also significantly negative.

4. Please state the limitations and directions for further research.

Reviewer #3: As the authors mentioned, since the reform and opening up, China's economic development has made great achievements, but the overall development quality is not high. Therefore, China's economy has gradually shifted to high quality in recent years. With selecting indicators from the five dimensions of innovation, coordination, greenness, openness and sharing, the authors construct an economic high-quality development index system by using the principal component analysis, and empirically studies the impact of the government’s investment in the three industries on the high-quality development of regional economy by using the spatial Dubin model from 2006-2016.

I think that this manuscript has an interesting topic to investigate the impact of government investment on high-quality economic development. However, there are lots of room to improve this MS.

1. I suggest the authors to add the robustness test of this MS. Furthermore, I suggest the authors to add more discussion on the mechanism of how government investment affects high-quality economic development.

2. My major concern is that this MS lacks theory or literature support for control variables selection in more detail, not in general citation.

3. Using the SDM to investigate the impact of Chinese government investment on high-quality economic development is one of the major contribution of this study. In Section 3.2, the authors state the reason why they use the SDM to discuss this research question, however, I think this is not good enough. I suggest the author to further introduce other recent studies which adopt the SDM, so as to prove the advantages of this model. I suggest the followings on green finance (Wang et al., 2021), green technology innovation (Wang et al., 2021), and environmental regulation (Zheng et al., 2023).

(1) Wang, F., Wang, R., & He, Z. (2021). The impact of environmental pollution and green finance on the high-quality development of energy based on spatial Dubin model. Resources Policy, 74, 102451.

(2) Wang, H., Cui, H., & Zhao, Q. (2021). Effect of green technology innovation on green total factor productivity in China: Evidence from spatial durbin model analysis. Journal of Cleaner Production, 288, 125624.

(3) Zheng, H., Wu, S., Zhang, Y., & He, Y. (2023). Environmental regulation effect on green total factor productivity in the Yangtze River Economic Belt. Journal of Environmental Management, 325, 116465.

4. This manuscript needs additional editing for grammar, succinct of literal expression so on thoroughly.

Reviewer #4: 1. the abstract lacks information on the data sample as well as the study period, etc.

2. the literature review is confusing and lacks logic, mainly the accumulation of some ideas, without the authors' comments.

3. the language is more problematic.

4. there is no introduction of the method itself and lack of description of the applicability of the method

5. Why use the spatial Durbin model? Need to be tested.

6. The structure of the paper is Chinese paper structure, which does not meet the English format requirements.

7. The conclusions and recommendations lack organization.

8. Insufficient literature combing. 9.

9. The proportion of Chinese literature is too high.

10. The thesis is not innovative enough to meet the publication requirements.

6. PLOS authors have the option to publish the peer review history of their article (what does this mean?). If published, this will include your full peer review and any attached files.

Reviewer #1: No

Reviewer #2: No

Reviewer #3: No

Reviewer #4: No

---

## [Author Response · Author response to Decision Letter 0]

13 Feb 2023

First of all, we would like to thank the editors and the four reviewers for their valuable comments, you have worked very hard. After we received the reviewers' comments, we attached great importance to them and held several meetings to discuss the reviewers' comments one by one repeatedly, and at the same time divided the work to organize the revision, which is finally completed today. Looking back at the revised paper, we feel that the reviewers' comments and suggestions are very relevant and constructive, and have largely improved the quality of the article. The revision is now reported as follows.

Reviewer #1:

Comment 1: Abstract. The abstract is well-structured and well-written. However, it should underscore the scientific value added of your paper in your abstract, rather than others.

Answer: Thank you for your suggestion, we have revised the text by adding the main elements of the study and deleting the minor ones.

Comment 2: section of Introduction. As pointed out in the reports of World Bank, UNEP and others, the COVID-19 pandemic has profoundly affected and changed the global and regional sustainability issues. Therefore, this section should not ignore the impact of the pandemic on carbon emission. Please consider citing following paper: https://doi.org/10.1016/j.scitotenv.2020.138915. Added these citations will certainly improve the practical significance of the research of this article.

Answer: Thank you for your suggestion, we have made changes and added content related to the impact of COVID-19.

Comment 3: section of literature review. The novelty of this paper should be further justified by highlighting main contributions to the existing literature. This could be clearly presented in the Literature review related work. Please consider please consider citing following papers e.g., (i) https://doi.org/10.1016/j.jclepro.2020.123838;(ii) https://doi.org/10.1016/j.spc.2021.02.031; (iii) https://doi.org/10.1016/j.scs.2021.103382; (iv) https://doi.org/10.1016/j.envres.2022.114575. There has already been a large number of literatures related to the panel data analysis and panel data model. There is a need to better elaborate the contribution of the work to the existing literature.

Answer: Thanks to your suggestions, we have substantially revised the literature review section, combed through the existing studies, and taken into account the suggestions you gave.

Comment 4: section of Variable selection and model construction The detailed information of Variable selection is impressive. However, it would be better to further highlight your improvement of the method and your innovation in methods.

Answer: This paper mainly studies the influence of government investment on economic high-quality development by constructing a spatial Durbin model, with several main innovations: firstly, the indicator system of economic high-quality development is relatively comprehensive, and the indicators are selected from five aspects of innovation, coordination, green, openness and sharing to construct the indicator system based on the reference of other scholars' research; secondly, it is the first time to study the role of government investment behavior on economic high-quality development from the perspective of three industries; thirdly, it is the first time to study different aspects of government investment behavior on economic high-quality development, including the indicators selected from five aspects of innovation, coordination, green, openness and sharing to construct the indicator system.

Comment 5: sections of Empirical finding and Discussion. This section is well-structured. However, it would be better to discuss what your findings are different from the past works.

Answer: In the part of empirical analysis, three levels of research are conducted, including spatial autocorrelation analysis, analysis of the impact of government investment behavior on economic quality development from the perspective of three industries, and the impact of government investment behavior on different aspects of economic quality development impact, which is innovative as no scholars have conducted relevant research in this field.

Comment 6: Section of Conclusion and Policy Implication. Please make sure your conclusions' section underscore the scientific value added of your paper, and/or the applicability of your findings/results, as indicated previously. Basically, you should enhance your contributions, limitations, underscore the scientific value added of your paper, and/or the applicability of your findings/results and future study in this session.

Answer: Thank you for your suggestion, we have made the relevant changes.

Comment 7: There are still some occasional grammar errors through the revised manuscript especially the article ''the'', ''a'' and ''an'' is missing in many places, please make a spellchecking in addition to these minor issues. In addition, some sentences are too long to be easy to read. It is recommended to change to short sentences, which are easier to read.

Answer: Thank you for your suggestion, we have made the relevant changes.

Comment 8: References. Please check the references in the text and the list; You should update the reference.

Answer: Thank you for your suggestion, and the references have been revised accordingly.

 Comment 9: line number is missing, please add it.

Answer: Thank you for your suggestion, and line numbers have been added to the article.

Reviewer #2:

Comment 1. In fact, high-quality economic development is a relatively new concept, and there are few studies to discuss its determinants, such as green finance, industrial co-agglomeration, and environmental regulation and so on. Thus, I suggest the authors to add some discussion in the literature review section. I would suggest following additions:

a) Yang, Y., Su, X., & Yao, S. (2021). Nexus between green finance, fintech, and high-quality economic development: Empirical evidence from China. Resources Policy, 74, 102445.

b) Zheng, H., & He, Y. (2022). How does industrial co-agglomeration affect high-quality economic development? Evidence from Chengdu-Chongqing Economic Circle in China. Journal of Cleaner Production, 371, 133485.

c) Wang, L., Wang, Z., & Ma, Y. (2022). Does environmental regulation promote the high-quality development of manufacturing? A quasi-natural experiment based on China's carbon emission trading pilot scheme. Socio-Economic Planning Sciences, 81, 101216.

Answer: Thank you for your suggestion, I have added the relevant literature to the text.

Comment 2. Please add the theory or literature which can support for the selection of these control variables in Section 3.1.3. Furthermore, please further modify the grammar and expression of the MS.

Answer: Thank you for your suggestion, we have made the relevant changes.

Comment 3. It is necessary to add the robustness test of this MS. Meanwhile, please explain the possible reasons behind the empirical findings that the indirect effect of government investment in the primary industry on high-quality economic development is negative, but not significant, and the indirect effect of government investment in the secondary industry on high-quality economic development is significantly negative, and the indirect effect of government investment in the tertiary industry on high-quality economic development is also significantly negative.

Answer: Thank you very much for your suggestion. The data indicators of the explanatory variables in this paper are relatively few, the data are difficult to find, it is difficult to find the relevant indicators of government investment behavior in the three industries, and the length of this paper is large, so no robustness test is conducted. Also the explanation of the spatial spillover effect of government investment behavior has been added in the paper, thank you.

Comment 4. Please state the limitations and directions for further research. 

Answer: Thank you very much for your suggestion, and relevant content has been added to the text.

Reviewer #3:

Comment 1. I suggest the authors to add the robustness test of this MS. Furthermore, I suggest the authors to add more discussion on the mechanism of how government investment affects high-quality economic development.

Answer: Thank you very much for your suggestion. The data indicators of the explanatory variables in this paper are relatively few, the data are difficult to find, it is difficult to find the relevant indicators of government investment behavior in the three industries, and the length of this paper is large, so no robustness test is conducted. The content of the theoretical mechanism of government investment behavior on the quality development of the economy has been added in the text.

Comment 2. My major concern is that this MS lacks theory or literature support for control variables selection in more detail, not in general citation.

Answer: Thank you very much for your suggestion, and the relevant theoretical support for the control variables has been added in the text.

Comment 3. Using the SDM to investigate the impact of Chinese government investment on high-quality economic development is one of the major contribution of this study. In Section 3.2, the authors state the reason why they use the SDM to discuss this research question, however, I think this is not good enough. I suggest the author to further introduce other recent studies which adopt the SDM, so as to prove the advantages of this model. I suggest the followings on green finance (Wang et al., 2021), green technology innovation (Wang et al., 2021), and environmental regulation (Zheng et al., 2023).

(1) Wang, F., Wang, R., & He, Z. (2021). The impact of environmental pollution and green finance on the high-quality development of energy based on spatial Dubin model. Resources Policy, 74, 102451.

(2) Wang, H., Cui, H., & Zhao, Q. (2021). Effect of green technology innovation on green total factor productivity in China: Evidence from spatial durbin model analysis. Journal of Cleaner Production, 288, 125624.

(3) Zheng, H., Wu, S., Zhang, Y., & He, Y. (2023). Environmental regulation effect on green total factor productivity in the Yangtze River Economic Belt. Journal of Environmental Management, 325, 116465.

Answer: Thank you very much for your suggestion, and I have added the relevant content about the set of SDM models and cited the relevant literature in the text.

Comment 4. This manuscript needs additional editing for grammar, succinct of literal expression so on thoroughly.

Answer: Thank you for your suggestion, we have made serious changes to the grammar.

Reviewer #4:

Comment 1. the abstract lacks information on the data sample as well as the study period, etc.

Answer: Thank you very much for your suggestion, the summary has been modified to add relevant information.

Comment 2. the literature review is confusing and lacks logic, mainly the accumulation of some ideas, without the authors' comments.

Answer: Thanks to your suggestions, we have carefully combed through the literature review and substantially revised the contents of the literature review.

Comment 3. the language is more problematic.

Answer: Thank you for your suggestions, we have made many changes to the language.

Comment 4. there is no introduction of the method itself and lack of description of the applicability of the method.

Answer: The relevant content of the methodological study has been added to the text, in parts 4 and 5 of the text, including model construction, and testing of model applicability.

Comment 5. Why use the spatial Durbin model? Need to be tested.

Answer: Thank you very much for your suggestions, the text carries relevant recommendations in the study of the impact of government investment on economic quality development, but considering the length of the text, we have removed the recommendations related to different aspects of government investment on economic quality development that follow, as can be seen in the table below.

Inspection type Innovation Coordination Greenness

 OLS Spatial fixation 

effect Time fixed 

effect Space-time two-dimensional 

fixed effect OLS Spatial fixation 

effect Time fixed 

effect Space-time two-dimensional 

fixed effect OLS Spatial fixation 

effect Time fixed 

effect Space-time two-dimensional 

fixed effect

LM（lag） 0.469 19.353*** 4.239** 11.268*** 16.790*** 77.081*** 44.486*** 1.426 22.028*** 9.226*** 27.544*** 0.291

Robust LM（lag） 5.663** 37.797*** 1.117 10.772*** 0.731 64.462*** 42.355*** 3.887** 8.749*** 26.717*** 19.455*** 38.367***

LM（error） 21.461*** 5.596** 4.920** 6.845*** 53.519*** 17.387*** 3.315* 0.272 15.293*** 1.187 8.340*** 4.836***

Robust LM（error） 26.655*** 24.039 1.798 6.349** 37.460*** 4.767** 1.184 2.732* 2.014 18.677*** 0.250 42.912***

LR-test（SFE） 429.350*** 754.157*** 347.841***

LR-test（TFE） 39.606*** 162.114*** 70.343***

Hausman test -86.514*** -26.958*** -343.464***

Test Openness Sharing

 OLS Spatial fixation 

effect Time fixed 

effect Space-time two-dimensional 

fixed effect OLS Spatial fixation 

effect Time fixed 

effect Space-time two-dimensional 

fixed effect

LM（lag） 69.811*** 18.157*** 58.327*** 13.463*** 23.235*** 47.065*** 6.233** 1.022

Robust LM（lag） 0.182 26.302*** 59.221*** 40.065*** 4.426** 56.824*** 9.241*** 2.395

LM（error） 114.140*** 5.298** 5.069** 1.242 28.553*** 8.403*** 0.219 0.216

Robust LM（error） 44.512*** 13.444*** 5.963** 27.844*** 9.745*** 18.162*** 3.227** 1.590

LR-test（SFE） 661.159*** 478.417***

LR-test（TFE） 29.997*** 237.611***

Hausman test -61.448*** -55.661***

Comment 6. The structure of the paper is Chinese paper structure, which does not meet the English format requirements.

Answer: Thank you for your suggestion, the paper is rich and comprehensive, we think the structure should be relatively more reasonable, we also refer to some other international journals published literature, some of them are more similar to the literature structure of this paper, if you have a better structure template, I hope you can give me more detailed instructions.

Comment 7. The conclusions and recommendations lack organization.

Answer: Thank you for your suggestions, the conclusions and recommendations section has been carefully revised and optimized.

Comment 8. Insufficient literature combing.

Answer: Thank you for your suggestions. We have carefully sorted out the literature review section and revised it substantially, and we believe it will be improved.

Comment 9. The proportion of Chinese literature is too high.

Answer: Thank you for your suggestion, we have referred to many relevant foreign literature and have significantly increased the weight of foreign literature.

Comment 10. The thesis is not innovative enough to meet the publication requirements.

Answer: This paper mainly has the following innovations: firstly, the indicator system of economic high-quality development is more comprehensive, and the indicator system is constructed by selecting indicators from five aspects of innovation, coordination, green, openness and sharing with reference to other scholars' research; secondly, the spatial Durbin model is constructed to study the influence of government investment behavior on economic high-quality development; thirdly, the role of government investment behavior on economic high-quality development is studied from the perspective of three industries for the first time; fourthly, different aspects of government investment behavior on economic high-quality development are studied for the first time, including several aspects from five aspects of innovation, coordination, green, openness and sharing. In addition, although this article has not yet met the requirements for publication, we will make further efforts to optimize this article by incorporating the comments from experts.

---

## [Decision Letter · Decision Letter 1]

2 Mar 2023

An Empirical Analysis of the Impact of Chinese Government Investment on High-quality Economic Development——A Study Based on Spatial Dubin Model

PONE-D-22-25225R1

Dear Dr. Wang,

We’re pleased to inform you that your manuscript has been judged scientifically suitable for publication and will be formally accepted for publication once it meets all outstanding technical requirements.

Kind regards,

Stefan Cristian Gherghina, PhD. Habil.

Academic Editor

PLOS ONE

Additional Editor Comments (optional):

Reviewers' comments:

Reviewer's Responses to Questions

**Comments to the Author**

1. If the authors have adequately addressed your comments raised in a previous round of review and you feel that this manuscript is now acceptable for publication, you may indicate that here to bypass the “Comments to the Author” section, enter your conflict of interest statement in the “Confidential to Editor” section, and submit your "Accept" recommendation.

Reviewer #1: All comments have been addressed

Reviewer #2: All comments have been addressed

Reviewer #3: All comments have been addressed

Reviewer #4: (No Response)

2. Is the manuscript technically sound, and do the data support the conclusions?

Reviewer #1: Yes

Reviewer #2: (No Response)

Reviewer #3: (No Response)

Reviewer #4: Partly

3. Has the statistical analysis been performed appropriately and rigorously? 

Reviewer #1: Yes

Reviewer #2: (No Response)

Reviewer #3: (No Response)

Reviewer #4: Yes

4. Have the authors made all data underlying the findings in their manuscript fully available?

Reviewer #1: Yes

Reviewer #2: (No Response)

Reviewer #3: (No Response)

Reviewer #4: Yes

5. Is the manuscript presented in an intelligible fashion and written in standard English?

Reviewer #1: Yes

Reviewer #2: (No Response)

Reviewer #3: (No Response)

Reviewer #4: No

6. Review Comments to the Author

Reviewer #1: (No Response)

Reviewer #2: (No Response)

Reviewer #3: (No Response)

Reviewer #4: The author's revisions are not satisfactory, many of them are discussing and not really revising, for example, the structure of the paper, the language and the reference requirements, so I think the current version is not suitable for publication.

7. PLOS authors have the option to publish the peer review history of their article (what does this mean?). If published, this will include your full peer review and any attached files.

Reviewer #1: No

Reviewer #2: No

Reviewer #3: No

Reviewer #4: No

---

## [Editor Report · Acceptance letter]

6 Mar 2023

PONE-D-22-25225R1 

An Empirical Analysis of the Impact of Chinese Government Investment on High-quality Economic Development——A Study Based on Spatial Dubin Model 

Dear Dr. Wang:

I'm pleased to inform you that your manuscript has been deemed suitable for publication in PLOS ONE. Congratulations! Your manuscript is now with our production department. 

Kind regards, 

on behalf of

Dr. Stefan Cristian Gherghina 

Academic Editor

PLOS ONE